# A Qualitative Review of Patient Feedback for the OPAT (Outpatient Antimicrobial Therapy) Service in Bristol

**DOI:** 10.3390/antibiotics13050420

**Published:** 2024-05-03

**Authors:** Shuchita Soni, Irasha Harding, Carys Jones, Sue Wade, Jenna Norton, Jennifer Siobhan Pollock

**Affiliations:** 1Department of Infectious Diseases, North Bristol NHS Trust, Southmead Hospital, Southmead Road, Bristol BS10 5NB, UK; 2Department of Microbiology, Bristol Royal Infirmary, Marlborough Street, Bristol BS1 3NU, UK; irasha.harding@uhbw.nhs.uk; 3Department of Pharmacy, Bristol Royal Infirmary, Marlborough Street, Bristol BS1 3NU, UK; carys.jones2@uhbw.nhs.uk (C.J.); sue.wade@uhbw.nhs.uk (S.W.); jenna.norton@uhbw.nhs.uk (J.N.); 4Sirona Care and Health CIC, Second Floor, Kingswood Civic Centre, High Street, Kingswood, Bristol BS15 9TR, UK; jennifer.pollock@uhbw.nhs.uk

**Keywords:** outpatient parenteral antimicrobial therapy (OPAT), patient feedback, patient satisfaction

## Abstract

Outpatient parenteral antimicrobial therapy (OPAT) aims to deliver intravenous antimicrobials to medically stable patients with complex infections outside of a hospital setting. There is good evidence to demonstrate the safety and efficacy of OPAT in the literature. Anecdotally, the feedback from patients has been positive, but only a few studies evaluate this topic in detail. The aim of this qualitative study was to examine patients’ experiences with and feedback on the OPAT service in Bristol, United Kingdom, which was established in 2021. A total of 92 patient feedback surveys were reviewed retrospectively, and thematic analysis was undertaken. Feedback from OPAT patients in our centre was overwhelmingly positive. The key themes identified were benefits to the patients, their friends, and family, and positive feedback about OPAT staff. The mean overall satisfaction score for OPAT was 9.6 out of 10. Areas to improve included communication between the OPAT and parent teams, improving OPAT capacity, and expansion of the service.

## 1. Introduction

OPAT (outpatient antimicrobial therapy) is a service that aims to deliver intravenous antimicrobials outside of a hospital setting. It is useful for patients who require parenteral therapy for the treatment of infection but are otherwise well enough to receive treatment outside a hospital. As there are often significant bed- and resource-related pressures within a hospital setting, the OPAT service can help deliver a vital service without patients having to be admitted as an inpatient [1].

OPAT services are a key component of the UK Health Security Agency Antimicrobial Stewardship Toolkit [2] and their availability enables early hospital discharge or admission avoidance for appropriately selected, medically stable patients. The service also incorporates principles of good antimicrobial stewardship and the infective needs of the patient and can consider an oral switch in a timely manner. The service can also monitor response to therapy and therefore shorten or increase the duration of the course of antibiotics [3].

The OPAT service in Bristol was set up in November 2021 [4]. It serves a population of 471,200 [5]. The two main hospital sites are North Bristol NHS Trust and University Hospital Bristol and Weston Trust. Sirona Care and Health covers the community and outpatient populations. Since 2021, the OPAT service has completed 593 patient episodes. In total, 37% of the admissions to OPAT were to facilitate early discharge and 63% were to avoid admission. Respiratory infections were the biggest syndromic group covered (18.6%). Following that were skin and subcutaneous tissue infections (8%) and then infections of the gastrointestinal tract (5%) [6]. These data are in keeping with national trends [7]. The service is set up so that specialist pharmacists and nursing staff visit patients in their homes and give feedback to the Infectious Diseases and Microbiology clinical teams. Some patients or their carers are taught to self-administer antibiotics, with their observations being monitored remotely, and a weekly nurse visit. Decisions are made on a weekly or twice weekly basis with a multi-disciplinary team meeting about continuing antibiotics, clinical progress, and if further investigations or referrals back to the parent team need to be made.

The benefits of OPAT have been well established in improving the patient experience, bringing the provision of healthcare closer to home, and efficiency and savings in healthcare. This would aid in faster recovery and the ability to maintain life at home [3]. Other advantages described have been limiting the amount of time spent in hospital and thus reducing the likelihood of contracting a hospital-acquired infection [8]. Antimicrobial stewardship is likely to be improved due to the continued involvement of specialized pharmacists and infection specialists. Some disadvantages have been highlighted in previous studies [9,10], such as the invasion and impact of privacy and disruption to home life from staff members visiting and administering the medication. Patients have mentioned that being unwell whilst being at home was tiring so they were unable to participate in home activities [9].

Few studies have examined patients’ satisfaction with OPAT services. It has been established from a previous study that the patient experience with the OPAT service is mostly positive. Overall, there were high levels of satisfaction due to patients being able to heal in their home environment, with less disruption to family life and greater comfort. There was also dissatisfaction due to nursing staff running late, limitations to patients’ lifestyles due to timings of the antibiotics, and anxiety associated with being home and not being well [11].

The importance of patient or service user feedback is well described in the literature [12]. Feedback collected could be used to assess performance of the service or to compare output and quality with other healthcare providers. Understanding the patient experience through the form of feedback is important in order to try and improve services. This should be used for quality improvement rather than a tick-box exercise. Feedback in the NHS is often collected as surveys in the “Friends and Family test”, national surveys, patient forums, and complaints systems. Opening up formal communication with patients and to encourage their involvement is helpful in strategic planning. The data collected should be utilized and analysed in a timely fashion [13].

This study explores patients’ experiences of the local OPAT service to identify positive indicators of the service and evaluate what areas could be improved.

## 2. Methods

Patient feedback forms were given to all patients at the end of their antimicrobial therapy with the OPAT service. This was during the timeframe of 4 March 2022 to 28 March 2023. The feedback form was designed based on the British Society of Antimicrobial Chemotherapy (BSAC) OPAT Patient Questionnaire [14]. Please see the Appendix A. Efforts to collect as many completed feedback forms were made by the OPAT team. There were no patients that were excluded from receiving or returning the forms. The forms were handed back in person or in a stamped addressed envelope that was provided. The completed forms have been uploaded onto a secure Microsoft Teams channel and have been anonymised. Only selected members of the team are given access to this channel. The majority of forms were sent back by post. The feedback forms have been analysed by thematic analysis to look for codes which were then themed [15]. The surveys were collated onto a master database and reviewed. The coding was conducted on a two-level basis. Initial open codes were highlighted and then these were further reduced to more focussed second-order codes. [15,16] A grounded theory approach—in that the themes emerge from the data and not be subject to the researcher’s personal bias or experiences—was used [17].

### 2.1. Study Settings

OPAT services covering the Bristol and surrounding areas were involved in collecting the data. The trusts that were involved were North Bristol NHS Trust, University Hospitals Bristol and Weston NHS Foundation Trust, and Sirona Health and Care Community Trust.

### 2.2. Sampling

All completed feedback forms for all OPAT patients were uploaded and included in the data. Duplicates were removed from the dataset.

## 3. Results

A total of 92 patients responded to the feedback survey. In total, 47% of the patients identified as female and 53% identified as male. The ages of the patients covered ranged from 16 years to 90 years. The greatest number of patients were in the 55 years to 84 years age group. There were 405 patient episodes, this makes for a survey response rate of 22.71%.

### 3.1. Numerical Scoring Systems

The mean overall satisfaction score was 9.6 out of a maximum score of 10. The lowest score identified was 4 and the highest was 10. The lowest score of 4 was given by one patient who was dissatisfied with the service due to the antibiotics felt to have been cut short prematurely and a feeling of lack of care by the OPAT teams. There appeared to be a lack of communication about the care of the patient between the parent team and the OPAT team, and the patient felt like information was lost between the teams. In total, 90% of patients said they were Extremely Likely to recommend the OPAT service to others, while 8.6% of patients said they were Likely to recommend the OPAT service to others, and 1% said they were Unlikely to recommend the service.

### 3.2. Codes and Themes from the Patient Survey

The key codes and emerging themes identified from the questionnaire are summarised in Table 1.

#### 3.2.1. Benefits to Patients

Being at home was a unanimous benefit to a significant proportion of the patients surveyed. Most patients found the healing environment of their own home helpful for comfort and to recover more quickly. Other benefits included better sleep, better for mental health, and freedom of movement and privacy compared to being in hospital.

“Able to get a good night’s sleep was way better for my mental health”.“It was amazing having the comfort of my own home and also getting reassurance and reviews from health professionals every day”.“Being at home helped me get better, quality and assurance was great and from a familiar team”.

Unexpectedly, it was noted that a few patients had stoically commented on the fact that they were able to give up their bed for a needier patient and that by using the OPAT service, this was able to be facilitated.

“Really good to be at home, saved me taking up a hospital bed that others might need”.

#### 3.2.2. Benefits to Friends and Family

The benefits to friends and family were important factors for our patients. This meant improvements in terms of travel and visiting times. Cost was a significant factor in terms of travel and parking. Childcare costs and responsibilities were important to the people visiting. Participating in family life had benefits in terms of returning to normality after a stay in hospital.

“I was able to be at home for my son’s 6th Birthday, it was hugely beneficial to me as I have two young children that I look after”.“Being out of hospital means less travel for my family, my wife is having treatment for cancer so this was particularly helpful”.“Less costs for us as a family due to petrol, parking and overall time spent in the car”.

#### 3.2.3. Staff Characteristics That Were Beneficial to Patients

The staff were well received by the patients under the OPAT team. Their friendliness, professionalism, and punctuality were key factors for patient satisfaction. Other human factors such as kindness, empathy, and reassurance were also important to our patient group. Communication between patients and the nursing teams was overall satisfactory. On a few occasions, patients found that the communication between the parent teams and OPAT teams had been difficult.

“Care was exemplary, great communication, great explanations on how to connect the IV”.“Friendly and everything was explained really well”.“Friendly, calm, patient and everything was professionally delivered”.

Key areas for improvement and dissatisfaction with the service are summarised in Table 2.

#### 3.2.4. Dissatisfaction with the Service

Although, the OPAT service generally received positive reviews, there were three patients whose reviews showed some dissatisfaction. Some of the main themes that emerged were lack of communication both with the patient and inter-agency (i.e., with parent teams and the OPAT teams). Dissatisfaction was associated with the OPAT services, focusing on the delivery of the antibiotics rather than review of the patient in a holistic approach. Some patients had queries about other medical conditions, and they were unable to obtain answers from the OPAT teams due to the focus on one area.

“There was no explanation of the plan to me, lack of communication”.“Confusing as to who was leading the antibiotic decisions”.“Poor communication from the teams with regards to the PICC line, the PICC line should have been taken out a while back”.“There was no mental health support”.“The number of agencies involved was confusing, staff a bit chaotic, poor communication amongst staff, once two nurses turned up at the same time!”“Lovely, kind and caring, but no one could answer my questions about other medical problems”.

#### 3.2.5. Characteristics of the Service That Were Positive

The service had positives that were highlighted in the patients’ feedback forms. Patients appreciated and felt empowered by being trained to be able to administer intravenous antibiotics for themselves and their loved ones. Patients appreciated staff punctuality and the use of new technology such as the elastomeric pump infusions.

“Loved the elastomeric pumps, they helped me get home soon”.“Great having the help at home and especially teaching my wife how to draw the IVs”.“They always came on time and called ahead if they were going to be late”.

#### 3.2.6. Improvements for the Service

Areas for improvement that were highlighted on the feedback forms were focused around increasing capacity for delivery of antibiotics in terms of frequency. The geographical catchment area covered by the service is also limited by the staffing capacity so that means some patients would not be eligible for the use of its service. Some mentioned a phone line for out-of-hours contact would be helpful. One patient mentioned that the nursing staff had not had the appropriate badges on their lanyard. Another patient would have liked a phone call prior to the nursing team coming to visit. Overall, most of the feedback for improvement was to continue the service in the same manner.

“24 h hotline for information and emergencies”.“Recruit more nurses into the team to expand the service”.“Better route planning across Bristol to save time/petrol and the environment”.

## 4. Discussion

The results of the study highlight that the OPAT service in Bristol is running in a manner that is satisfactory to the patient experience, and the feedback obtained was overwhelmingly positive. The overall benefits of receiving intravenous therapy at home has improved patients’ health and mental health as well as reduced disruption to family life. Friends and family have benefited from reduced traveling times as well as costs for childcare and parking fees. The gaps in communication between clinical teams and patients have been highlighted as areas to improve on in the future. These gaps can be attributed to the lack of continuity on medical rotas and difficulties in smoothly accessing the parent team. The feedback also suggested that a more unified or holistic approach in caring for the patient would be beneficial. The nursing teams are unable to answer all concerns due to time limitations and gaps in knowledge or training. Their focus is mostly on delivering the antibiotic therapy. Some feedback on aspects of care such as pain or intolerances to antibiotics is fed back to the OPAT medical team and adjustments are made. However, more could be done to review the entirety of the patient’s medical care. One option would be to establish a clinic so the patients can be reviewed medically or a more virtual ward-based approach. UHBW will be trialling this method by reviewing select patients on a weekly basis in the infectious diseases clinic.

## 5. Limitations

Although the study looks at all the feedback collected since the conception of the service, there are some limitations in the way the data may have been collected. Limitations of this qualitative study were that it was a snapshot of patients’ experience during their time with the OPAT team. There was a low response rate to filling out the feedback form. The feedback forms may show a positive selection bias because patients who are more likely to fill out forms are those that have had a positive experience. The patients that have filled out the forms may have an element of recall bias as they were handed the forms at the end of therapy. The experience at the end of therapy may have been different from the experience during the antibiotic course [12]. There were some gaps in the collection of the forms due to loss of forms and non-return of forms. There may be some inadvertent exclusion of patients who have physical or mental disabilities that would prevent the paper forms from being filled out and returned.

In order to be able to capture a greater amount of feedback, there could be options in the future to use QR codes and online feedback as well as paper feedback. Patients could also be given the feedback link or forms early on in their therapy, so they are able to accurately record their experience. A weekly check-in could also be an option and a real-time way of collecting data and feedback. This would make responding to the feedback more effective and individualized to patient needs.

## 6. The Future of the OPAT Service in Bristol

The OPAT service has proven to be an invaluable service for Bristol and the surrounding areas. It has reduced the number of bed days needed for antibiotic therapy by 7494 bed days. It has also enriched and improved the quality of life for the patients enrolled and their families and carers. The service is still evolving to meet the needs of the population and the hospitals. The team is also expanding with nursing staff learning new skills and increasing the capacity of the service being provided. There are skills that are specific to one site which will be transferable to train the staff at other sites, e.g., intravenous therapy training for other members of the family and carers.

Elastomeric pumps have revolutionised the way antibiotics are delivered and the ability to maintain the time above the minimum inhibitory concentration (MIC) by improving the pharmacokinetic/pharmacodynamic (PK/PD) index. Continuous infusions help with this [18]. There have been some ongoing supply chain issues with the delivery of these pumps. In the future, better and more stable procurement contracts will help eliminate the need for waiting as an inpatient for the installation of the pumps.

Another limiting factor for the progression of patients migrating to the OPAT service is the insertion of central intravenous access. Ideally, for longer term antibiotic schedules, patients should have midlines or centrally inserted peripheral vascular access. This is to prevent the need to change intravascular access devices and to reduce the risk of bloodstream infections [9,19]. Currently, there is a limited number of practitioners that can insert these, so this remains a factor in preventing patients from being discharged in a timely fashion.

The service is likely to expand with the increasing needs of the local population and limitations in hospital resources. Currently, the service is able to offer twice-a-day delivery of antibiotics; however, with increasing staff training and retention, it is hoped that the number of times a day that antibiotics can be delivered can be improved. Elastomeric pumps have also helped with increasing the delivery of antibiotics.

In the future the service is likely to follow a COPAT (complex outpatient antibiotic therapy) service model [20]. This would be where a multi-disciplinary team would supervise patients on long-term oral antimicrobial therapy as well as intravenous therapy. In a COPAT model, there is more scope to be able to monitor toxicity, drug–drug interactions, and to be able to monitor patients for treatment failure on oral antibiotics. This movement has become more prevalent since the OVIVA [21] and POET [22] trials, whereby intravenous therapy is not always the only option. Early oral step down with close monitoring of clinical signs and biochemical tests can be beneficial to patients and cost saving for hospitals [20,23]. A COPAT model would help increase the number of patients that could be cared for outside of a hospital setting, as well as improve antimicrobial stewardship, reduce antibiotic related adverse events, and improve overall satisfaction for patients [20,24,25]. The increase in capacity for patients would be because of the earlier oral switch of antibiotics, which would mean less traveling for the nursing staff, thereby freeing up time. Earlier intravenous to oral switch would also help achieve better antimicrobial stewardship and reduce the chance of intravenous catheter associated infections [19]. Caution should be exercised within a COPAT model as there may not be enough robust data to support earlier oral switch in some deep-seated infections; however, with close monitoring and the involvement of a multi-disciplinary team, this system seems likely to be beneficial in the future [25].

## 7. Conclusions

The OPAT service has been a positive service for patients in Bristol. Albeit in its infancy, it has shown to be a service that is flexible and can accommodate significant numbers of patients. This allows cost-saving measures for the hospitals as well as overall benefits to patients. The feedback reviewed has been positive and has highlighted areas to focus on in the future. The balance between human factors and communication by the nursing teams, and the interactions between the OPAT team and parent teams, is paramount in keeping the quality of the service to such a high level. This study highlights the positive impact that the service has on patients in the Bristol area as well as the parts of the service that can be improved upon. The future of the service is resource dependent, but it is hopeful that it will be able to be expanded and a more holistic COPAT-based approach be adopted.

This review of the feedback has highlighted areas for future research. In-depth data could be collected by doing focus groups and interviews with patients to explore their ideas and expectations of the service in detail. A review of the feedback form is underway. The form is to be made more inclusive and accessible, as well as to improve ease of collection and response rate from patients. The addition of an online version of feedback forms or QR codes would be beneficial in moving away from paper-based methods of feedback. A review of feedback collected would help improve feedback collection methods. As the importance of feedback and quality improvement has been highlighted previously, feedback collection and analysis should be paramount in shaping future services in an evidence-based fashion.

## Figures and Tables

**Table 1 antibiotics-13-00420-t001:** Codes and themes identified from the patient survey.

Benefits to the Patient	Benefits to the Patient’s Family	Other
Ability to recover in their own home	Reduced time and cost spent on visiting patient in hospital	Kindness and professionalism demonstrated by the OPAT team staff
Better sleep than in a hospital environment	Ability to spend time with friends and family	Giving up a hospital bed to someone who might need it more
Greater freedom of movement	Less disruption of family life	

**Table 2 antibiotics-13-00420-t002:** Areas of dissatisfaction with the OPAT service and suggested areas for improvement.

Communication	Other Areas of Dissatisfaction	Areas for Improvement
Unsatisfactory communication with the patient	Lack of mental health support	Increasing nursing staff to improve OPAT capacity
Unsatisfactory communication between the OPAT and parent teams	Need for a more holistic approach	Extend the geographic area visited by the OPAT team
		Out-of-hours hotline for patients

## Data Availability

Please contact us directly for data reviewed. All questionnaires are stored and available if requested.

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
