# Peer review of "A Qualitative Review of Patient Feedback for the OPAT (Outpatient Antimicrobial Therapy) Service in Bristol"

_antibiotics, 2024, doi:10.3390/antibiotics13050420_

Round 1

Reviewer 1 Report (Previous Reviewer 2)

Comments and Suggestions for Authors

I find the manuscript well-written, as it describes all the necessary details about this qualitative study, addressing all the methodological limitations and the most important results.  

Author Response

thanks for your comments

Reviewer 2 Report (Previous Reviewer 1)

Comments and Suggestions for Authors

The article is well-written and interesting

Author Response

thanks for your comments

This manuscript is a resubmission of an earlier submission. The following is a list of the peer review reports and author responses from that submission.

Round 1

Reviewer 1 Report

Comments and Suggestions for Authors

I suggest adding some references to previous studies discussing the importance of patient feedback to improve the quality of healthcare services. This could further support your argument about the relevance of the study. You could also add some references about the benefits of this type of treatment, for example, the reduction of claims due to HAI and of costs for compensation; on this topic, I’ll suggest the following references:

-          Medico-Legal Aspects of Hospital-Acquired Infections: 5-Years of Judgements of the Civil Court of Rome. Treglia, M., Pallocci, M., Passalacqua, P., ...Cisterna, A.M., Marsella, L.T. Healthcare (Switzerland), 2022, 10(7), 1336

Briefly describe the process of selecting patients to participate in the feedback analysis. For example, what criteria were used to include or exclude patients?

In the "Numerical scoring systems" paragraph, you could briefly discuss the reasons behind the lowest score of 4 to provide a complete context for the range of scores received.

In the "The Future of the OPAT service in Bristol" section, you could further explore the idea of a COPAT (Complex Outpatient Antibiotic Therapy) approach and explain why it might be advantageous compared to the current OPAT model.

In conclusion, you could emphasize the importance of the recommendations for service improvement and how these might impact patients' future experiences. You could also consider briefly mentioning any further research or studies that could stem from this work.

Author Response

Dear Colleague, thank you for taking the time to review our paper. The comments have been helpful in improving the outcomes of the paper. Please see the points highlighted below. 

I suggest adding some references to previous studies discussing the importance of patient feedback to improve the quality of healthcare services. This could further support your argument about the relevance of the study. You could also add some references about the benefits of this type of treatment, for example, the reduction of claims due to HAI and of costs for compensation; on this topic, I’ll suggest the following references:

  •          Medico-Legal Aspects of Hospital-Acquired Infections: 5-Years of Judgements of the Civil Court of Rome. Treglia, M., Pallocci, M., Passalacqua, P., ...Cisterna, A.M., Marsella, L.T. Healthcare (Switzerland), 2022, 10(7), 1336

Dear Colleague thank you for your suggestion for the inclusion of the above mentioned reference and about the incidence of HAI (Hospital Acquired infections) and how OPAT can facilitate in reducing these. I have included this reference in our report.

Briefly describe the process of selecting patients to participate in the feedback analysis. For example, what criteria were used to include or exclude patients?

Thank you for this comment, as mentioned in the methods section there are no patients that were excluded from receiving or returning the feedback forms. 

In the "Numerical scoring systems" paragraph, you could briefly discuss the reasons behind the lowest score of 4 to provide a complete context for the range of scores received.

Thank you for the comment. I have given an explanation as to why a low score was given by one patient. I hope the explanation about their care is adequate and is understandable as to why they scored the service low. 

In the "The Future of the OPAT service in Bristol" section, you could further explore the idea of a COPAT (Complex Outpatient Antibiotic Therapy) approach and explain why it might be advantageous compared to the current OPAT model.

Thank you for this comment,  I have included a whole paragraph as to how a adopting a COPAT model will likely be advantageous to the service. This would improve capacity, free up staff time and improve antimicrobial stewardship. 

In conclusion, you could emphasize the importance of the recommendations for service improvement and how these might impact patients' future experiences. You could also consider briefly mentioning any further research or studies that could stem from this work.

Thank you for this comment, please see the comments about how we are trialling a face to face clinic and also improving the feedback form to make it more accessible and improve pick up rates. This will help us be more reactive in the way we address issues highlighted in the feedback forms to improve the service overall. 

Please do contact me if any further clarifications are needed. 

Yours Sincerely, 

Dr Shuchita Soni 

Reviewer 2 Report

Comments and Suggestions for Authors

This study aimed to examine patients' experiences and feedback on the OPAT service in Bristol.

However, I have found several flaws with the study design and the presented results.

1. There is no reported survey response rate, as it is only stated that "92 patients responded to the feedback survey". As "the OPAT service has completed 593 number of patient episodes", does it mean this is 92/593 response rate?

2. Both these sentences in the Limitations section raise a  red flag for me: "There were some gaps in the collection of the forms due to loss of forms and non-return of forms. There may be some exclusion of patients who have physical or mental disabilities that would prevent the paper forms from being filled out." There are no including, non-including and excluding criteria in the Methods. There is no straightforward procedure for the survey collection. It seems that the survey's paper and non-paper forms (MS Teams) were collected - but why and how is unclear.

3. There is a strong selective bias in how the survey has been conducted. It is more designed as an OPAT advert than a scientific study. The ridiculously high number of satisfaction scores highlights this: "9.6 out of a maximum score of 10".

4. Minor remarks: a) There are no authors' affiliations in the manuscript; b) Section 5.1. is unnecessary and out of place; c) References need reformatting.  

Author Response

Dear Colleague, thank you for taking the time to review our article, the comments have been really helpful in refining our article. Please see below the answers to your points. 

  1. There is no reported survey response rate, as it is only stated that "92 patients responded to the feedback survey". As "the OPAT service has completed 593 number of patient episodes", does it mean this is 92/593 response rate? - Thank you for this comment, please see the updated feedback survey response rate as mentioned in the methods section.  Of the 405 patient episodes there were 92 patient feedback forms collected. This makes the feedback form response rate of 22.7%

2. Both these sentences in the Limitations section raise a  red flag for me: "There were some gaps in the collection of the forms due to loss of forms and non-return of forms. There may be some exclusion of patients who have physical or mental disabilities that would prevent the paper forms from being filled out." There are no including, non-including and excluding criteria in the Methods. There is no straightforward procedure for the survey collection. It seems that the survey's paper and non-paper forms (MS Teams) were collected - but why and how is unclear.

Thank you for this comment, as updated in the methods section for inclusion and exclusion factors. The feedback form was handed out to all patients who had been under the OPAT team. The limiting factor for collating a greater number of feedback forms would have been the fact that patients hadn't returned them. Due to the fact that it was a postal form, people may have found it difficult to find the time or were unable to post the form back. We have addressed this as an issue for future area of improvement. 

3. There is a strong selective bias in how the survey has been conducted. It is more designed as an OPAT advert than a scientific study. The ridiculously high number of satisfaction scores highlights this: "9.6 out of a maximum score of 10".

We have looked at the feedback data and forms again and most people have marked satisfaction with the service as 10/10. There is selection bias noted as possibly people who are more likely to give feedback are those that are satisfied with the service.  We have discussed this in the limitations section. 

4. Minor remarks: a) There are no authors' affiliations in the manuscript; b) Section 5.1. is unnecessary and out of place; c) References need reformatting.  

Thank you for these remarks - these have been updated and amended. 

Thank you for your overall input on this paper. I hope that the comments have been addressed and I hope you are able to re-review the paper to see if it is suitable publishing. 

Yours Sincerely, 

Dr Shuchita Soni